# Adsorption of Anthocyanins by Cation and Anion Exchange Resins with Aromatic and Aliphatic Polymer Matrices

**DOI:** 10.3390/ijms21217874

**Published:** 2020-10-23

**Authors:** Natalia Pismenskaya, Veronika Sarapulova, Anastasia Klevtsova, Sergey Mikhaylin, Laurent Bazinet

**Affiliations:** 1Kuban State University, 149 Stavropolskaya st., 350040 Krasnodar, Russia; vsarapulova@gmail.com (V.S.); nastya-k1314@yandex.com (A.K.); 2Department of Food Sciences, Institute of Nutrition and Functional Foods (INAF), Laboratory of Food Processing and ElectroMembrane Process (LTAPEM), University Laval, Québec, QC G1V, Canada; Sergey.Mikhaylin@fsaa.ulaval.ca (S.M.); Laurent.bazinet@fsaa.ulaval.ca (L.B.)

**Keywords:** anthocyanins, structure, adsorption kinetics, adsorption isotherms, ion-exchange resin, Donnan exclusion, electrostatic interactions

## Abstract

This study examines the mechanisms of adsorption of anthocyanins from model aqueous solutions at pH values of 3, 6, and 9 by ion-exchange resins making the main component of heterogeneous ion-exchange membranes. This is the first report demonstrating that the pH of the internal solution of a KU-2-8 aromatic cation-exchange resin is 2-3 units lower than the pH of the external bathing anthocyanin-containing solution, and the pH of the internal solution of some anion-exchange resins with an aromatic (AV-17-8, AV-17-2P) or aliphatic (EDE-10P) matrix is 2–4 units higher than the pH of the external solution. This pH shift is caused by the Donnan exclusion of hydroxyl ions (in the KU-2-8 resin) or protons (in the AV-17-8, AV-17-2P, and EDE-10P resins). The most significant pH shift is observed for the EDE-10P resin, which has the highest ion-exchange capacity causing the highest Donnan exclusion. Due to the pH shift, the electric charge of anthocyanin inside an ion-exchange resin differs from its charge in the external solution. At pH 6, the external solution contains uncharged anthocyanin molecules. However, in the AV-17-8 and AV-17-2P resins, the anthocyanins are present as singly charged anions, while in the EDE-10P resin, they are in the form of doubly charged anions. Due to the electrostatic interactions of these anions with the positively charged fixed groups of anion-exchange resins, the adsorption capacities of AV-17-8, AV-17-2P, and EDE-10P were higher than expected. It was established that the electrostatic interactions of anthocyanins with the charged fixed groups increase the adsorption capacity of the aromatic resin by a factor of 1.8–2.5 compared to the adsorption caused by the π–π (stacking) interactions. These results provide new insights into the fouling mechanism of ion-exchange materials by polyphenols; they can help develop strategies for membrane cleaning and for extracting anthocyanins from juices and wine using ion-exchange resins and membranes.

## 1. Introduction

Anthocyanins are classified as polyphenols, and are composed of carbohydrate moieties (glycoside) connected to anthocyanidins (aglycone) via the oxygen of the hydroxyl group in the 3-position of anthocyanidin (Figure 1). Anthocyanidin differs in substituents, R, which can be -H, or -OH, or -OCH_3_ groups. The carbohydrate moiety of anthocyanins most often consists of glucose but may also consist of other mono- and disaccharides, for example, rhamnose, arabinose, and galactose. It is known [1] that the structure and electric charge of all anthocyanins depend on the pH of the medium (Figure 1), due to participation of their structural elements in protonation/deprotonation reactions: in an acidic medium, anthocyanins acquire a positive charge and act as cations while in alkaline conditions, they acquire a negative charge and become anions. 

Anthocyanins are natural antioxidants [2], used to prevent and treat cancer [3] and diabetes [4], as well as cardiovascular [5] and neurodegenerative diseases [6]. They are also actively used as food dyes [7,8], colorimetric indicators of food freshness [9], cosmetics, and dietary supplements for detoxification of the human body and increasing its immunity [10]. The unique properties of these substances have caused an avalanche-like growth in related publications. Indeed, in 2000, Scopus indexed 314 publications on mainly the medical properties of these substances, but since 2015, the number of publications has exceeded 2000 per year. The main focus of this research has shifted to the preservation of anthocyanins and other polyphenols in juices and wines during their production [11,12,13] as well as to anthocyanins extraction from natural sources (primarily from winemaking waste [4]); in processing berries, fruits, vegetables [14,15,16,17,18,19,20]; or in discoloration of food [21,22] and waste water [23,24,25]. 

Traditional extraction methods for removing anthocyanins use acidified water–alcohol mixtures or other organic solvents [26,27]. However, these methods do not always provide the desired degree of purity: the recovery liquid media contains anthocyanins in small quantities and/or as a part of a mixture with many other components. Using sorption methods [28,29,30] in combination with membrane (nanofiltration, ultrafiltration, electrodialysis) technologies [14,30,31] provides higher efficiency and allows selective separation of anthocyanins and other polyphenols from multicomponent natural mixtures. Several reviews exist on the sorption and membrane methods used to extract and purify anthocyanins [15,29,32,33]. Activated carbon and nonpolar or weakly polar porous polymer resins are the most common adsorbents [15,34,35,36,37]. Special attention is paid to the specific surface area, pore, and granule sizes of these adsorbents [38], as well as the temperature of the adsorption process [39,40]. A number of researchers also highlighted that the nature of the adsorbent can significantly affect the performance of the process [15,41,42]. Enhanced adsorption can be achieved by π–π (stacking) and Van der Waals interactions between the aromatic rings of polyphenols and the aromatic polymer matrix of adsorbents [42,43]. Consequently, increasing the pH above *pK_a_* (*K_a_* is the phenol dissociation constant) is a common method for regenerating nonpolar or weakly polar adsorbents [44,45,46].

In recent years, technologies using ion-exchange resins (IERs) [15,47,48,49,50,51] or ion-exchange membranes (a type of flat resin) [18,19,20] for the extraction and purification of phenols, and anthocyanins, in particular, have been increasingly adopted. These materials differ from adsorption resins only by having polar fixed groups. In the case of anion-exchange resins, these groups have a positive electric charge while in cation-exchange resins, these groups are negatively charged. Furthermore, it was established that using ion-exchange resins increases the selectivity and efficiency of phenol [50,52,53,54], polyphenol, and anthocyanin [13,47,48,54,55,56,57,58,59] adsorption compared to nonpolar macroporous resins and activated carbon. IERs also allow polyphenols to be separated from other components of processed liquid media [60,61]. Moreover, purification of anthocyanins increases their antioxidant activity [62] but traditional eluents for the extraction of phenolic compounds from nonpolar resins are often unsuitable in the case of IER [54]. 

Current investigations suggest that phenols enter into π–π (stacking) and Van der Waals interactions with ion-exchange materials [54,55], in a similar way as with nonpolar resins. Therefore, the sorption increases when switching a polymer matrix of IERs and membranes from aliphatic (e.g., polyacrylamide) to aromatic (polystyrene) [63]. In contrast to nonpolar adsorbents, hydrogen bonds are formed between fixed IER groups and phenol hydroxyl groups [64,65]. The fixed groups of ion-exchange materials then form electrostatic interactions with phenols, when the latter are in ionic form in the solution [42,54,55,56,57,58]. Furthermore, the proportion of ionic forms depends on the dissociation constants *pK_a_* of the polar groups of polyphenols and the pH of the solution [13,47,48,54,55,56,57,58,59]. However, there is no consensus in the literature concerning the relationship between the pH of a processed solution and the adsorption capacity of an IER. For example, using cation-exchange resins to extract biogenic amines and sulfates from wine at pH 3.5 [66] or protons from fruit juices [67] does not lead to adsorption of anthocyanins [65] or this adsorption is negligible [66] despite the fact that anthocyanins are cations in these liquid media. Ghafari et al. [54] found that phenol is equally adsorbed by anion-exchange materials at both pH 6 and 11 of the external solution. Since the *pK_a_* of phenol is 9.9, these authors [54] concluded that phenol was unlikely to be deprotonated at pH 6, so electrostatic interactions between phenol and the anion-exchange material were unlikely. Thus, different adsorption mechanisms prevail at pH 6 and 11: formation of hydrogen bonds at pH 6 and electrostatic interactions at pH 11.

We performed this study to clarify the main mechanisms underlying the adsorption of anthocyanins by IERs from aqueous solutions with different (3, 6, and 9) pH values. The study includes the identification of the structure of anthocyanins in the external solution and within the IERs having aromatic or aliphatic matrices and differing in porosity and exchange capacity. We will analyze the kinetics and equilibrium of the anthocyanin sorption by the resins, taking into account the structure and electric charge of anthocyanins. 

## 2. Results and Discussion

### 2.1. Effect of pH on Anthocyanin Structure in External Solutions and Ion-Exchange Resins

#### 2.1.1. In External Solutions

External solutions were prepared from an aqueous anthocyanins mixture. According to the manufacturer (DOO “Frutarom Etol”, Skofja vas, Slovenia), it was extracted from grape pulp. This extract has pH 2.95 ± 0.05 and, according to high-performance liquid chromatography (HPLC) analysis, mainly contains the anthocyanins cyanidin and peonidine (Table 1). Their structures at pH 3 are shown in Figure 2. The pH of external solutions was adjusted to 3 ± 0.05, 6 ± 0.05, and 9 ± 0.05 by adding NaOH.

Figure 3 shows the fingerprint region (400–1800 cm^−1^) of the infrared (IR) spectra of the anthocyanin solutions at pH 3, 6, and 9. These spectra had three characteristic bands in the region of 1440–1610 cm^−1^ assigned to the skeletal vibrations of C=C groups of benzene rings [68,69]. Previous studies [70,71] indicated the peak at 1610 cm^−1^ corresponded to aromatic compounds bearing –OH groups in the ortho position. The wide intense band between 1590 and 1610 cm^−1^ corresponded to the stretching vibrations of C–C bonds and in-plane bending vibrations of the benzene ring of the catechin group (ring B). The shoulder, or peak, at 1515 cm^−1^ was due to the stretching vibrations of C–C bonds and in-plane vibrations of the benzene ring of the resorcinol fragment (ring A) [70]. In addition, this lone peak confirmed the presence of simple anthocyanins in the test solution since compounds based on anthocyanins of various degrees of polymerization gave two sharp peaks at 1535 and 1520 cm^−1^ [69]. The absorption at 1448–1444 cm^−1^ was caused by the asymmetric in-plane bending of –CH_3_ [72,73], which corresponds to the structure of peonidin. Furthermore, the phenyl nucleus (C=C) absorbed in the same spectral region while –CH_2_– deformed [72,73,74]. 

The asymmetric and symmetric vibrations of the C–O–C bond of the pyran ring manifested in the ranges of 1260–1285 cm^−1^ and 1025–1198 cm^−1^, respectively [69]. The band between 920 and 930 cm^−1^ corresponded to the skeletal vibration involving α-1,4-glycosidic bond (C–O–C) [75]. The catechin fragment with two neighboring O–H groups also corresponded to the bands at 1280 and 1245 cm^−1^ [76]. The out-of-plane and in-plane bending vibrations of C–H in the aromatic ring corresponded to the 670–900 cm^−1^ region [77]. The peak between 735 and 770 cm^−1^ was typical of the bending vibrations of the C–H bond of orthosubstituted aromatic compounds [78]. In the 400–750 cm^−1^ range, the peaks were caused by out-of-plane and in-plane bending vibrations of the C–O–H bond of the catechin fragment, with the out-of-plane vibrations being assigned to higher wave numbers than the in-plane vibrations [76].

The structural features of anthocyanins were present since, at pH 3, the 1720 cm^−1^ peak of the IR spectrum corresponded to the positively charged C=O^+^ group [78]. At pH 6 and 9, this peak disappeared, and another peak whose intensity decreased with increasing pH appeared in the region of 1400–1410 cm^−1^. These changes in the spectra indicated the transformation of the –C=O group of the chromenylium cycle (pH 3) to the –O– group (pH 6 and 9), which had no electric charge [79]. The spectra obtained for the anthocyanin solutions at pH 3 and 6 produced similar peaks in the 860 cm^−1^ and 760 cm^−1^ regions. At pH 9, these peaks shift to 867 cm^−1^ and 775 cm^−1^. In addition, this spectrum had peaks at 850 cm^−1^ and 610 cm^−1^, which are absent at lower pH values. Increasing the pH of the external solution transformed the IR spectra in the range of 670–900 cm^−1^ indicating changes in the side groups of the chromenylium cycle and benzene rings [80]. At pH 6, colorless pseudo bases with non-protonated –OH groups prevailed in the anthocyanin mixtures, yet some in the quinoid form, which has one of the phenolic hydroxyl groups in the deprotonated state; –O^−^, was also found. At pH 9, quinoid forms predominated in the anthocyanin mixtures; however, a certain number of phenolates were detected which had not one but two deprotonated –O^−^ groups. In addition, some chalcone forms were present. The chalcones formed due to quinoid hydrolysis, which was accompanied by a rupture of the chromene cycle.

The IR-spectroscopy data agree with the colorimetric scale (Figure 4) obtained for different pH values of the anthocyanin mixture. According to this scale, acidic solutions (1 < pH < 3) are bright red, which is typical of the pyrylium salt of anthocyanins and is an indicator of the dominance of flavylium cations in the solution. As the pH increased from 3 to 7, the color gradually changed to purple due to an increase in the proportion of the quinoid form, which has no electric charge. In the range of pH 7 to 9, shades of blue typical of the anthocyanin quinoid form, which is a singly charged anion, became increasingly predominant. Around pH 10, shades of yellow and green prevailed, indicating the presence of doubly charged anions in the solution. A small part of this was the chalconoid form with the ruptured chromene cycle. At higher pH values, chalcone anions became the dominant anthocyanin forms producing solutions that became more and more yellow. This scale confirmed the well-known [1] relationship between the structure and the color of anthocyanins (Figure 1). However, the shift to the blue and green colors corresponding to singly charged anions of anthocyanins is observed in a slightly more alkaline region than indicated by Ribéreau et al [1]. The cause for this difference was that each of the anthocyanins in the model mixture had their own constants of protonation/deprotonation reactions of radicals. Therefore, the color shifts are less abrupt compared to separated substances or substances alone.

#### 2.1.2. In Ion-Exchange Resins

The following IERs with typical ion-exchange polymer matrices and fixed groups (Table 2) were studied. The anion-exchange resin EDE-10P (manufacturer: PJSC “Uralchimplast”, Russia) was made by polycondensation of polyethylene polyamines (PEPAs) with epichlorohydrin (ECH) [81]. This resin had an aliphatic matrix and contained primary, secondary, and tertiary amines and a small number of quaternary ammonium bases as fixed groups. The other resins (manufacturers: TD “Ural Chemical Company”, Russia; JSC “Azot”, Ukraine) were made by suspension polymerization.

Their polymer matrixes consisted of polystyrene (PS) regularly crosslinked with divinylbenzene (DVB) at a fraction of 2% (AV-17-2P) or 8% (AV-17-8, KU-2-8) [86]. The aromatic anion-exchange resins contained mainly quaternary amines and small amounts of secondary and tertiary amines as fixed groups. The cation exchange resin KU-2-8 contained sulfonic groups. The true density and total ion-exchange capacities of the water-swollen IERs are shown in Table 2. The diameter of the resin beads is in the range of 0.3–0.5 mm.

Figure 5 shows the IR-spectra of samples of the aliphatic anion-exchange resin EDE-10P. Prior to the experiment, one resin was equilibrated with deionized water (EDE-10P_W_), and the other with an anthocyanin solution at pH 3 (EDE-10P_Ant_^pH3^). These and other IR spectra of IER were interpreted using the data from several previous studies [78,87,88,89]. The weak double band typical of the bending vibrations of N–H appeared in the 1614 to 1633 cm^−1^ region of the EDE-10P_W_ IR spectra, indicating the presence of secondary and primary amines in the resin. Numerous peaks characteristic of C–N stretching occurred between 1020 and 1174 cm^−1^, also indicating the presence of weakly basic fixed amino groups in the anion exchange resin [54,90].

Furthermore, peaks appeared in the following regions of the IR-spectra of EDE-10P_W_: (1) 2963 to 2844 cm^−1^, corresponding to the asymmetric and symmetric stretching vibrations of –CH_2_ groups; (2) 1459 cm^−1^, corresponding to the bending vibrations of C–H groups; and (3) 1030 to 1020 cm^−1^ corresponding to stretching vibrations of the C–OH bond. It is well known [91] that these are the main groups constituting the EDE-10P resin. For the EDE-10P_Ant_^pH3^ sample, the peaks at 1614 and 1633 cm^−1^ overlapped with intense maxima typical of C=C bonds in the benzene rings of anthocyanins. The intensity of the other peaks typical of EDE-10P_W_ was enhanced due to the adsorption of anthocyanins, which included glycosides with –CH_2_, C–H, and C–OH groups. In the EDE-10P_Ant_^pH3^ IR spectrum it was not possible to distinguish the peaks at 1722 and 1519 cm^−1^, which are characteristic of the positively charged flavylium cation form of anthocyanins; however, there were peaks at 870 and 780 cm^−1^, which are characteristic of anthocyanin with non-protonated –OH groups and some number of deprotonated phenolic hydroxyl groups. As shown in Figure 5, the EDE-10P_Ant_^pH6^ and EDE-10P_Ant_^pH9^ IR spectra differed from the EDE-10P_Ant_^pH3^ spectrum in a shift of these peaks to the 867 and 775 cm^−1^ regions with increased intensity, as well as the peaks appearing at 827 and 611 cm^−1^, indicating the increase of deprotonated phenolic hydroxyl groups.

IR spectroscopic data agreed with the observed change in EDE-10P resin color depending on the pH of the external solution. Indeed, this resin in the chloride form and equilibrated with deionized colorless water was orange in color (Figure 6a). The EDE-10P_Ant_^pH3^ resin sample in contact with the carmine red solution of anthocyanins (pH 3) acquired a purple-brown hue (Figure 6b). The EDE-10P_Ant_^pH6^ resin sample equilibrated with the purple solution of anthocyanins (pH 6) turned a yellow-green (khaki) color (Figure 6c). The EDE-10P_Ant_^pH9^ resin sample equilibrated with the gray-blue solution of anthocyanins (pH 9) and became yellow-brown (Figure 6d). The bright color of the pristine resin made it difficult to interpret the data obtained; however, this was achieved by taking into account the rules of color science [92]. The registered colors were produced by mixing red-orange hues (Figure 6a) with blue (Figure 6b), with green (Figure 6c), and with yellow (Figure 6d). This indicated that single-charged anions of the quinoid form of anthocyanins (EDE-10P_Ant_^pH3^), the blue isomer of the doubly charged chalcone anion (EDE-10P_Ant_^pH6^), or the yellow doubly charged isomer of the chalcone anion ruptured the chromene cycle prevailing in the resin.

IR spectroscopy of anthocyanins adsorbed by the aromatic resins AV-17-2P, AV-17-8, and KU-2-8 was an even greater challenge than for the aliphatic EDE-10P resin due to the very similar chemical structure between the IER polymer matrix and anthocyanin. Analysis of these spectra (data not shown) showed an increase in the absorption intensity at the wavelengths associated with the presence of aromatic –C=C– bonds or characteristic of glucosides. These intense extended peaks shifted to the ultraviolet region, indicating the presence of the π–π (stacking) interactions of anthocyanin aromatic rings with poly(styrene-divinylbenzene) IER matrix [87], and overlapped with the wavelengths (1722 and 1519 cm^−1^) assigned to the positively charged flavilium cation. Thus, the colors that the aromatic resins have acquired gave more information about the structure and electric charge of the anthocyanins adsorbed by them in comparison with the IR spectra.

In water and aqueous solutions of strong electrolytes, the aromatic AV-17-2P resin was white in color (Figure 7a). Therefore, color changes associated with conversions in the anthocyanin structure were not distorted by overlapping color hues. The AV-17-2P_Ant_^pH3^ resin sample in contact with the carmine red solution of anthocyanins (pH 3) acquired a purple color (Figure 7b) corresponding to pH 6.0 on the colorimetric scale. This meant that anthocyanins were in the form of flavilium cations in the solution, but in the resin changed to the purple phenolate of the quinoid form, which has no electric charge. The AV-17-2P_Ant_^pH6^ resin sample equilibrated with the pH 6 anthocyanin solution, where anthocyanins were present as the purple phenolate of the quinoid form bearing no electric charge, acquiring a gray-blue color (Figure 7c). This color corresponded to a pH value of 9 on the colorimetric scale, where most of the anthocyanins were transformed into blue quinoid anions with a charge of 1^−^. In the pH 9 solution, anthocyanins were present as a mixture of neutral molecules and singly charged anions. The AV-17-2P_Ant_^pH9^ resin sample equilibrated with this solution became greenish-yellow (Figure 7d). This color corresponded to a pH value of 11 according to the colorimetric scale, indicating that the resin contained blue-green and yellow isomers of the chalcone anion bearing an electric charge of 2^−^.

While the colorimetric analysis of anthocyanin charge in the AV-17-8 resin (data not shown) was similar to those obtained for AV-17-2P, the behavior of the cation-exchange resin equilibrated with the anthocyanin solutions differs from the anion-exchange resin. The color of the KU-2-8_Ant_^pH3^ sample turned dark red, corresponding to pH 1.65 on the colorimetric scale (Figure 8b). The KU-2-8_Ant_^pH6^ resin sample became a purple-pink color, corresponding to pH 4 on the colorimetric scale (Figure 8c). The KU-2-8_Ant_^pH9^ resin sample turned brownish-orange in color (Figure 8d). These results indicate that at pH 3 of the external solution, anthocyanins in the cation-exchange resin are singly charged cations. At pH 6 and 9, they are a mixture of colorless pseudo bases with no electric charge and a small number of anthocyanin cations, or a small number of anthocyanin anions.

Thus, the data presented in Figure 6, Figure 7 and Figure 8 indicate that the structure of anthocyanins inside ion-exchange resins differs markedly from the structure of these substances in external solutions. The structure in anion-exchange resins (Figure 6 and Figure 7) corresponds to pH values that are 2–3 units higher than in external solutions of anthocyanins. Conversely, the structure of anthocyanins in a cation exchange resin (Figure 8) corresponds to pH values that are 2–3 units lower than in external solutions of anthocyanins. This phenomenon is similar to mechanisms already reported for ion-exchange membrane fouling by red wine components, primarily anthocyanins [93]. Hence, the reason for the pH shift of the internal solution of ion-exchange membranes and resins is the well-known [85] Donnan (electrostatic) exclusion of coions. Figure 9 illustrates the Donnan effect in ion-exchange materials if coions are the products of protonation/deprotonation reactions involving water molecules. The scheme is presented when the pH of the external solution is 6 and where the electric charge of anthocyanins is zero (Ant^0^). In the case of anion-exchange resin (Figure 9a), the coions are protons with the same electric charge as positively charged fixed groups of anion-exchange resins. These protons are excluded from anion-exchange resins due to the Donnan effect. As a result, the pH of the anion-exchange resin internal solution increases. Once in this alkaline medium, electrically neutral anthocyanin molecules are transformed into anions (Ant^−^) due to the deprotonation of phenolic hydroxyl groups. The higher the concentration of fixed groups, the more intense the Donnan exclusion of protons [85]. That is why anthocyanins in EDE-10P resin, which is characterized by the highest exchange capacity (Table 2), have a more negative charge than in resins AV-17-2P and AV-17-8.

Conversely, protons going from anion-exchange resin into the external solution lowers its pH, which can transform some of the neutral anthocyanin molecules into cations (Ant^+^). In cation-exchange resins (Figure 9b), the coions are hydroxyl ions and Donnan exclusion of hydroxyl ions decreases the pH of the resin’s internal solution. In this acidic medium, neutral anthocyanin molecules (quinoidal anhydrobases) are transformed into another neutral form (carbinol pseudobases) or modified into cations. At the same time, hydroxyl ions entering the external solution participate in the deprotonation reactions of neutral anthocyanin molecules, generating a certain number of anthocyanin anions in the external solution. Note that these anthocyanin anions (for cation-exchange resin) and cations (for anion-exchange resin) in the solution at the surface of an ion-exchange material can decrease the adsorption capacity of a resin if they are not quickly and consistently removed to the solution volume, for example, by mixing. Similar to anthocyanins, phenols, amino acids, carboxylic acids, polybasic organic acids, and other ampholytic substances that are components of juices and wines [1] participate in protonation/deprotonation reactions with water, also affecting the pH of the internal solution of ion-exchange materials [93,94,95,96]. This particular effect is one of the reasons not previously discussed that produces differences in anthocyanin adsorption between single solutions and mixtures, for example, with amino acids [55,56].

The pH values of external and internal solutions of IERs, as well as the electric charges of anthocyanins in these mediums, are summarized in Table 3. Knowing these charges and pH will help interpret the kinetic adsorption isotherms presented in the next section. 

### 2.2. Effect of the External Solution pH on Anthocyanin Adsorption by Ion-Exchange Resins

Figure 10 shows the kinetics of anthocyanin adsorption by anion- and cation-exchange resins from the aqueous solutions at pH 3, 6, and 9. First, let us analyze the kinetics of the adsorption of anthocyanins by aromatic resins, which are produced from a copolymer of polystyrene with divinylbenzene. It should be noted that the fixed groups of KU-2-8 are negatively charged sulfonic groups, whereas positively charged AV-17-8 are mainly quaternary ammonium bases. The AV-17-2P resin has the same fixed groups and less crosslinking (therefore, larger pores) than AV-17-8. The ion exchange capacity of these resins in a swollen state grows in the following order: AV-17-2P < AV-17-8 < KU-2-8.

Anthocyanin flavilium cations, Ant^+^, which are coions for anion-exchange resin, dominate in the acidic external solution (pH 3) of the aromatic resins AV-17-2P and AV-17-8. However, when moving into the internal solution of AV-17-2P and AV-17-8, they are transformed into the uncharged molecules, Ant^0^ (Table 1). The adsorption of the molecular form of anthocyanins to these resins is mainly accomplished through π–π (stacking) interactions of the anthocyanin aromatic rings with each other and with the IER aromatic matrix [41,42,43]. The macroporous AV-17-2P resin has larger pores [84] than the AV-17-8 resin, allowing large anthocyanin molecules to penetrate more easily into the AV-17-2P resin and delivering its higher adsorption capacity and the adsorption factor of anthocyanins (Table 1). At pH 3, the AV-17-8 anion-exchange resin adsorbs the least anthocyanins while the KU-2-8 cation-exchange resin (Figure 10a) adsorbs the most. The differences in behavior of these aromatic resins with respect to anthocyanins are mainly determined by the degree of the electrostatic interactions of anthocyanins and fixed groups. Indeed, the KU-2-8 cation-exchange resin equilibrated with the external solution at pH 3 contains Ant^+^ cations (Table 3). This cation is the counterion of the negatively charged sulfonic groups of KU-2-8. Thus, the high adsorption capacity of KU-2-8 for anthocyanins is due to at least two types of interactions: (1) the electrostatic interactions between the counterions Ant^+^ and sulfonic fixed groups of KU-2-8 and (2) the π–π (stacking) interactions of the aromatic matrix of the resin with the benzene rings of anthocyanins. Given that the resins AV-17-8 and KU-2-8 have the same aromatic matrix, it can be assumed that the more than 54% higher adsorption capacity and the coefficient of adsorption concentrating of anthocyanins (Table 3) of the cation-exchange resin KU-2-8 are mainly due to the electrostatic interactions (Figure 10a).

When aqueous solutions of anthocyanins have pH values of 6, the behavior of the resins changes radically. The adsorption capacity (Figure 10b) of the KU-2-8 cation-exchange resin and the adsorption factor of anthocyanins (Table 1) become respectively 2.1 times less than that of aromatic anion-exchange resin and 2.1 times less than its own adsorption capacity for a solution at pH 3 (Figure 10a). Indeed, in the internal solution of KU-2-8, the pH is 4 (Table 3), so anthocyanins lose their electrical charge. Therefore, electrostatic interactions of the cation-exchange resin KU-2-8 with Ant^0^ are absent, and π–π (stacking) interactions become the main mechanism of their adsorption. On the contrary, when penetrating into AV-17-2P and AV-17-8, anthocyanins acquire a negative charge, Ant^−^ (Table 3), and, in addition to π–π (stacking) interactions with the ion-exchange matrix, electrostatic interactions take part with the positively charged fixed groups of these anion-exchange resins. Note that the increase in adsorption capacity (Figure 10a,b) and absorption factor (Table 3) of anthocyanins during their conversion from Ant^0^ to Ant^+^ (KU-2-8) or Ant^−^ (AV-17-2P and AV-17-8 ) increases in the same order as the ion exchange capacity: AV-17-2P < AV-17-8 < KU-2-8. 

Finally, if the pH of the external solution is equal to 9 (Figure 10c), the anthocyanins inside the cation-exchange resin generally remain non-charged molecules (Table 3). Therefore, the kinetic isotherms of the adsorption of anthocyanins by the KU-2-8 resin are close to the curves obtained with the external solution at pH 6 (Figure 10b). Under the same conditions (the external solution at pH 9), inside the anion-exchange resins anthocyanins acquire an electrical charge 2^−^, Ant^2-^ (Table 3). Nevertheless, the adsorption capacity of these resins (Figure 10c) and the adsorption factor of anthocyanins (Table 3) remain the same as in the case of the neutral pH values of the external solution (Figure 10b, Table 3). A similar effect was noted by other authors [42,54,97,98] when studying the adsorption of phenols. This effect is probably caused by deprotonation of the weakly basic secondary and tertiary amines in an alkaline medium [99,100]. As a result, secondary and tertiary amines that AV-17-8 and AV-17-2P resins contain as impurities [81] no longer participate in electrostatic interactions with negatively charged anthocyanins at high pH values of the internal solution of these resins (Table 3).

The relatively high content of primary, secondary, and tertiary amines predetermines the peculiarities of the behavior of EDE-10P in comparison with other studied resins. The highest EDE-10P ion-exchange capacity provides the greatest pH shift to the alkaline range in comparison with AV-17-8 and AV-17-2P resins (Таble 3). That is why EDE-10P contains not only Ant^0^ molecules, but also a certain amount of Ant^−^ anions (counterions), whereas they are in the form of cations Ant^+^ in the external solution (pH 3). This means that electrostatic interactions of anthocyanins with positively charged fixed groups of this resin are not dominant. Moreover, the resin has an aliphatic ion-exchange matrix, which excludes the possibility of its π–π (stacking) interactions with the aromatic rings of anthocyanins [101]. However, the adsorption capacity (Figure 10a) and the adsorption factor (Table 3) of the EDE-10P are higher than those for aromatic ion exchange resins, in which the π–π (stacking) interaction (AV-17-8 and AV-17-2P) or π–π (stacking) and electrostatic interactions (KU-2-8) of anthocyanins are the dominant adsorption mechanisms. This phenomenon is apparently caused by the formation of hydrogen bonds between primary, secondary, and tertiary amines and hydroxyl groups of anthocyanin molecules. The latter type of interaction is more significant in the case of weakly basic fixed groups, which prevail in EDE-10P, compared to quaternary ammonium bases of AV-17-8 and AV-17-2P resins [54,102,103].

Increasing the pH of the external anthocyanin solution to 6 and 9 increases the pH of EDE-10P internal solutions to 11 and 12 (Table 3). Consequently, the fraction of Ant^2-^ anions increased in this resin. At the same time, the fraction of deprotonated weakly basic fixed groups grew, reducing the ability of these groups to participate in hydrogen bonds and electrostatic interactions with anthocyanins. The combined effect of both phenomena leads to a noticeable decrease in the adsorption capacity of EDE-10P when the pH of the external solutions was at 6 (Figure 10b) and especially at 9 (Figure 10c). In the case of pH 9, the negative effect of deprotonation of weakly basic fixed groups is so great that the adsorption capacity (Figure 10c) and the adsorption factor (Table 3) of the EDE-10P resin become close to KU-2-8, which do not enter into electrostatic interactions with Ant^0^ molecules and Ant^−^ anions. The maximum on the adsorption isotherms of EDE-10P (Figure 10b,c) is apparently caused by the fact that the processes of deprotonation of fixed groups, destruction of hydrogen bonds, and diffusion of “released” anthocyanins from the resin back into the external solution take time. 

These results clarify some of the discussion points mentioned in the Introduction. In particular, this is an equally high adsorption of phenols and polyphenols by strong basic anion exchange materials from neutral and alkaline solutions, which were observed despite the fact that in these external solutions they are uncharged molecules and anions, respectively [42,54,97,98]. The key moment to explain this phenomenon is the pH shift of the internal solution of anion-exchange materials to the alkaline range due to the Donnan exclusion of protons—the products of the protonation–deprotonation reactions. Thus, in a neutral external solution, these substances are uncharged molecules, but getting into the anion-exchange resin, they become anions. This provides electrostatic interactions with the resin at the same level as in the case of alkaline external solution in which phenol and polyphenols are initially in the deprotonated state. These results are important for the practice of extracting anthocyanins, because they justify the use of anion exchange resins under conditions in which anthocyanins retain a higher biological activity [104,105].

As to cation-exchange resins, the well-known fact is confirmed: acidification of the external solution enhances the adsorption of anthocyanins due to the significant increase in electrostatic interactions.

### 2.3. Effect of the Concentration of Anthocyanins in External Solutions on Their Adsorption by Resins

Figure 11 shows the kinetic dependences of the adsorption of anthocyanins by aromatic resins, which demonstrated the maximum (KU-2-8) and minimum (AV-17-8) adsorption capacity in the case of a solution of anthocyanins with pH 3. These curves were obtained for the concentrations of anthocyanins in an initial external solution (pH 3) from 10 to 100 mg dm^–3^ and used to obtain equilibrium adsorption isotherms indicated by the sub-index E (Figure 12). The results of processing these isotherms using Langmuir, Freundlich, and Brunauer–Emmett–Teller (BET) models [106,107] and linear regression methods (Section 3.3.4.) are summarized in Table 4. 

Attention should be paid to the fact that the kinetic isotherms of adsorption of the cation exchange resin KU-2-8 from a solution at pH 3 have two distinct plateaus (Figure 11a). However, plateau I becomes less and less noticeable with an increase in external solution pH (Figure 10c), that is, under conditions when anthocyanin cations become uncharged molecules. The presence of two plateaus when anthocyanins in the resin are cations suggests that at the beginning, the adsorption of anthocyanins occurs mainly due to weak π–π (stacking) interactions with the KU-2-8 aromatic matrix. Diffusion of large species of anthocyanins into the resin, their participation in protonation–deprotonation reactions, and reverse diffusion of protolysis products take from 60 (*C_S0_* = 10 mg dm^−3^) to 30 (*C_S0_* = 100 mg dm^−3^) minutes. A further increase in the concentration of adsorbate in KU-2-8 is mainly provided by electrostatic interactions of anthocyanin counterions with fixed resin groups. The ratio *C_IER_^II^*/*C_IER_^I^*, where *C_IER_^I^* and *C_IER_^II^* are the anthocyanin concentrations in the resin, determined from plateaus I and II, gives an idea of the contribution of these electrostatic interactions to the total adsorption capacity of the cation exchange resin. The ratio is equal to 1.4 (*C_S0_* = 10 mg dm^−3^), 2.8 (*C_S0_* = 50 mg dm^−3^), and 3.4 (*C_S0_* = 100 mg dm^−3^), and, as reported in references [33,50,102], increases with increasing anthocyanin concentration in the solution. This growth is apparently caused by phenomena coupled with electrostatic interactions, which require further study.

In the case of the AV-17-8 anion-exchange resin, a weakly expressed plateau I is recorded for neutral and alkaline external solutions (Figure 10b,c), when anthocyanins inside the resin acquire a negative electric charge (Table 3) and become counterions for resin positively charged fixed groups. On the contrary, there is no plateau I if the external solution has a pH of 3 (Figure 11b) since the anthocyanins inside the resin lose their electrical charge. Apparently, in this case, the formation of hydrogen bonds and other interactions between the hydroxyl groups of anthocyanins and the ammonium groups of AV-17-8 partially compensate the electrostatic adsorbent–adsorbate interactions. It can be assumed that the intra-diffusion kinetics limit this type of interaction to a much lesser extent. This predetermines the weak manifestation of plateau I on the kinetic curves of adsorption even in the case when π–π (stacking) interactions and the formation of hydrogen bonds are supplemented by electrostatic adsorbent–adsorbate interactions. 

Both equilibrium isotherms (Figure 12) have borderline shape between the type V and VI isotherms; concave initial sections correspond to isotherm type III according to the IUPAC classification [108]. Such a concave type of isotherm generally corresponds to adsorption with relatively weak adsorbent–adsorbate interactions. It is believed that this weakness causes a small capacity at the beginning, but if at least one molecule is adsorbed, adsorbate–adsorbate interactions further enhance adsorption of other molecules. Indeed, this initial section I (up to point B) of equilibrium isotherms is best described using the Freundlich model, which characterizes adsorption with a non-uniform distribution of adsorption affinities over adsorbent surface [109]. The values of the adsorption constants found using the Freundlich equation (Table 4) are in good agreement with studies on the adsorption of various polyphenols by ion-exchange resins. These studies are summarized, for example, in reviews [15,109]. The Langmuir equation, used to describe section I, gives a noticeably lower value of the coefficient of determination, *R*^2^ (Table 4); moreover, the found amount of adsorbate corresponding to complete monolayer adsorption, *q_m_*, and the Langmuir isotherm constant, *K_L_*, have negative values; that is, they have no physical meaning. This result is expected because this model takes into account monolayer adsorption that only occurs at fixed identical and equivalent sites with no lateral interactions or steric hindrance between an adsorbent and adsorbate.

Point B on the “knee” of equilibrium isotherms characterizes the transition from monomolecular to polymolecular adsorption (section II). An indirect confirmation of the polymolecular adsorption of anthocyanins is the identification of dimers and trimers of these substances in ion-exchange membranes [110] that had a chemical structure similar to the studied resins and put in contact with anthocyanin-containing solutions. Among the reasons for such polymolecular adsorption are the π–π (stacking) interaction of the aromatic rings of polyphenols with each other [29,50] or with substances with the opposite electric charge at a given pH [111]. We assume that in our case the main reason for polymolecular adsorption is the π–π (stacking) interactions between already adsorbed anthocyanins and those in the pores of the resin or in the external solution. Really, the external anthocyanin solution does not contain polyphenols (Table 1). In addition, the pores of the studied resins (and membranes [100]) are too small for the penetration of polyphenols.

Processing section II of adsorption isotherms using the Freundlich model gives a high coefficient of determination and allows us to conclude that the adsorption constants increase by a two order of magnitude if the equilibrium concentration of anthocyanins in the external solution becomes greater than 30 mg dm^−3^. At the same time, the BET model, which is most often used to describe the polymolecular adsorption of phenols [50,112,113] and polyphenols [65,114] by ion-exchange resins, gives a rather low coefficient of determination values (Table 2). The use of other (Frenkel–Halsey–Hill, MacMillan–Teller) models of polymolecular adsorption [109] known to us also does not give satisfactory results. Apparently, this is caused by the too complex nature of the interactions between anthocyanins and the studied ion-exchange resins.

## 3. Materials and Methods 

### 3.1. Ion-Exchange Resins

Prior to experiments, the resins were immersed in a saturated (300 g dm^−3^) NaCl solution for 24 h and then rinsed with small volumes of deionized water. The rinsing was completed when the electrical conductivity and pH of water, equilibrated with the resin for one hour, differed from less than 5% from that of initial deionized water.

### 3.2. Solutions

In experiments, we used a natural dye—anthocyanin—which was purchased from Frutarom Etol d.o.o. (Škofja Vas, Slovenia).

A colorimetric scale of aqueous (40 mg dm^−3^) solutions of anthocyanins in the pH 1 to 12 range was obtained using the buffer solutions listed in Table 5.

All reagents (CJSC «Vekton», Saint Petersburg, Russia) were of analytical grade. Deionized water with an electrical conductivity of 1.5 μS cm^−1^ and a pH of 5.5 was used for all the experiments.

### 3.3. Methods

#### 3.3.1. The pH Differential Spectrophotometric Method

The pH differential method [34,115] using a LEKI SS 2107 spectrophotometer (Leki Instruments, Finland) was applied to determine the total concentration of anthocyanins in solution. Two 0.5 cm^3^ samples were taken from the solution and their volume increased to 10 cm^3^ by adding pH 1 or 4.5 buffer solutions. After 15 minutes, the optical density of the resulting solutions was determined at wavelengths of 520 and 700 nm. The optical density, *A*, of anthocyanins was calculated using the equation: $A={\left(A_{520}{-A}_{700}\right)}_{\mathrm{pH}\;1}-{\left(A_{520}{-A}_{700}\right)}_{\mathrm{pH}\;4.5}$. The total concentration of anthocyanins in the solution, *C_s_* (mg dm^−3^), was expressed as equivalent of cyanidin-3-glucoside using the following expression:(1)$$C_{S}=\frac{AMV_{1}}{V_{2}\varepsilon l}\;\times 10^{3},$$
where *M* is the molecular weight of cyanidin-3-glucoside (449.2 g mol^−1^); *V_1_* is the volume of the volumetric flask used to dilute the sample (cm^3^); *V_2_* is the sample volume taken for analysis, (cm^3^); *ε* is the molar extinction coefficient of cyanidin-3-glucoside (26,900 [mol cm dm^−3^] ^−1^); and *l* is the optical path length of a cuvette (1 cm).

#### 3.3.2. High-Performance Liquid Chromatography (HPLC) Analysis

The composition of the anthocyanins extract from grape pulp used in the experiments was determined by HPLC [116] using the Agilent 1100 HPLC system (Agilent Technologies, Santa Clara, CA, USA) and a Luna 2 × 250 mm C18 column (Phenomenex, Torrance, CA, USA).

#### 3.3.3. Fourier Transform Infrared (FTIR) Spectroscopy and Optic Analysis

Two methods were used to determine the structures of anthocyanins in the solutions and resins. FTIR spectra were obtained using a Vertex-70 (Bruker Optics, Germany) and the ATR (attenuated total reflection) accessory in the wavelength range of 4000–400 cm^−1^ with 4 cm^−1^ spectral resolution and 32 scans. The intensity of the spectra was normalized by subtracting the recorded baseline using OPUS™ software. Prior to measurements, the solutions of anthocyanins were dried out by evaporating the moisture at 40 °C. The resin samples were equilibrated with the external solutions of anthocyanins at the given pH values. After that, the IERs were removed from the solutions, ground, and dried to constant weight at 50 °C.

In addition to the information provided by the IR spectra, the structures of anthocyanins adsorbed by the IER were assessed by comparing the colors of the resin using the colorimetric scale of the anthocyanin solutions. This scale was obtained using buffer solutions (Table 5) with pH values from 1 to 12 and the known data on the conversion of their structure as a function of pH [1] (Figure 1). Optical images of the anthocyanin solutions and IERs before and after their contact with the anthocyanin solutions were obtained at constant luminous intensity (ensured by the constant electric power 14 W ± 5% consumed by the light source) and an optical path length of 225 mm using a SOPTOP CX40M optical microscope (China) with a set of 5× objective and a digital eyepiece camera.

#### 3.3.4. Kinetics and Equilibrium Isotherms of the Anthocyanins Adsorption

The kinetics of the anthocyanins’ adsorption by the ion-exchange resins were determined using the constant volume method. Samples of the swollen ion-exchange resin weighing 1.0 ± 0.01 g were placed in 10 flasks with ground stoppers. A standard volume (20 cm^3^) of the anthocyanin solution of a given concentration and pH was added and the mixture was shaken at 80 min^−1^ frequency using an EcoPribor PE-0034 shaker. Samples (0.5 cm^3^) from each flask were taken after a predetermined time (from 10 to 180 min) to determine the concentration of anthocyanins and the pH of the solution after contact with the resin.

The total concentration of anthocyanins in an ion-exchange resin (mg g^–1^) was determined using the following equation:(2)$$C_{\;IER}=\frac{\left(C_{S0}-C_{St}\right)V_{s}}{V_{IER}},$$
where *C_S0_* and *C_St_* are total concentrations of anthocyanins in the initial and predetermined time interval of contact of the external solution with the resin (mg g^−1^); *V_IER_* = *m*/*ρ* is the volume (cm^3^); *m* and *ρ* are the mass (g) and true density (g cm^−3^) of the swollen resin, respectively; and *V_s_* is the volume of solution in contact with the resin (in cm^3^). All experiments were carried out under isothermal conditions at a temperature of 25 ± 1 °C.

In this study, Langmuir, Freundlich, and Brunauer–Emmett–Teller (BET) models [106,107] were used to describe the adsorption equilibrium between the liquid phase and the adsorbent. Isotherm equations of these models are presented in Table 6. 

## 4. Conclusions

The focus of this study was to characterize mechanisms of anthocyanin adsorption by ion-exchange resins from model aqueous anthocyanin-containing solutions at different pH values. 

Using optical color indication and IR spectroscopy, it is shown that the pH of the internal solution is 2–3 units lower in the case of an aromatic cation-exchange resin KU-2-8, and 2-4 units higher in the case of anion-exchange resins with aromatic (AV-17-8, AV-17-2P) and aliphatic (EDE-10P) matrix compared to the pH of the external anthocyanin solution. This effect is caused by Donnan exclusion of hydroxyl ions (KU-2-8) or protons (AV-17-8, AV-17-2P, EDE-10P), which are products of anthocyanins protolysis reactions. The Donnan exclusion increases with the increasing ion-exchange capacity of the resin. Thus, the most noticeable pH shift was observed in the EDE-10P resin, which had the highest ion-exchange capacity.

In the external solutions with pH 3, 6, and 9, anthocyanins are cations, uncharged molecules, and singly charged anions, respectively. Due to the shift in pH, the electrical charge of anthocyanins inside the ion exchange resins differs from their charge in the external solution. As a result, in KU-2-8 internal solution anthocyanins remain cations (pH 3 external solution) or become uncharged molecules (pH 6 and 9 external solutions). In the AV-17-8 and AV-17-2P internal solutions, they become uncharged molecules (pH 3 external solution), singly charged (pH 6 external solution), or doubly charged (pH 9 external solution) anions. If anthocyanins inside the resin are counterions, their electrostatic interactions with fixed groups essentially strengthen the adsorption caused by π–π (stacking) interactions of the aromatic rings of the adsorbate and the adsorbent. Our estimates show that in the case of KU-2-8 (pH 3 external solution), electrostatic interactions increase the maximum adsorption capacity of the resin by 1.4-3.4 times compared with that which can be achieved due to π–π (stacking) and other possible interactions. In the investigated range of anthocyanin concentrations in the external solution (from 10 to 100 mg dm^−3^), the contribution of electrostatic interactions to the adsorption capacity of the resin depends on the contact time of the IER with the external solution and increases with an increase in the adsorbate concentration in it.

At a given concentration of anthocyanins in an external solution (40 mg dm^−3^), the maximum adsorption capacity of KU-2-8 in an acidic external solution increases by 2 times compared to neutral and alkaline solutions. In the case of the anion-exchange resins AV-17-8 and AV-17-2P, the adsorption capacity increases by only 1.4 and 1.5 times when passing from acidic to neutral and alkaline external solutions. The decrease in the effect of electrostatic interactions of the adsorbent–adsorbate in comparison with the cation-exchange resin, apparently, is caused by the partial deprotonation of weakly basic fixed groups of anion-exchange resins when the internal solution is strongly alkaline (pH 11–12). This deprotonation is tantamount to a partial loss of the resin’s ion exchange capacity.

The aliphatic resin EDE-10P, when in an external solution at pH 3, contains a mixture of anions and uncharged anthocyanin molecules. The high adsorption capacity of this resin, which exceeds the one observed in the case of KU-2-8, is apparently due to a combination of electrostatic interactions between the adsorbate and the adsorbent, as well as the formation of a large number of hydrogen bonds between the hydroxyl groups of anthocyanins and EDE-10P primary, secondary, and tertiary amines. Deprotonation of these groups in a highly alkaline medium of the internal solution (pH 6 or 9 external solution) negates the gain from the conversion of anthocyanins into doubly charged anions–counterions. As a result, the maximum adsorption capacity of EDE-10P is observed in the case of a weakly acidic external solution (pH 3). 

Polymolecular adsorption in the systems under study is clearly manifested if the concentration of anthocyanins in the external equilibrium solution exceeds 30 mg dm^−3^.

The knowledge gained in this study not only expands our understanding of the adsorption mechanisms of anthocyanins by aromatic and aliphatic IERs but also has practical value. Indeed, to ensure the maximum adsorption capacity of anion-exchange resins, it is better to not use alkaline (as it is often practiced), but neutral or even slightly acidic aqueous solutions of anthocyanins. This conclusion is also important because in an alkaline environment, anthocyanins can undergo partial destruction and lose their biological activity [107,108]. This new knowledge can also help to develop the strategies of resins and membrane cleaning as well as the strategies for extracting anthocyanins from wine and juice waste using ion-exchange resins and membranes.

## Figures and Tables

**Figure 1 ijms-21-07874-f001:**
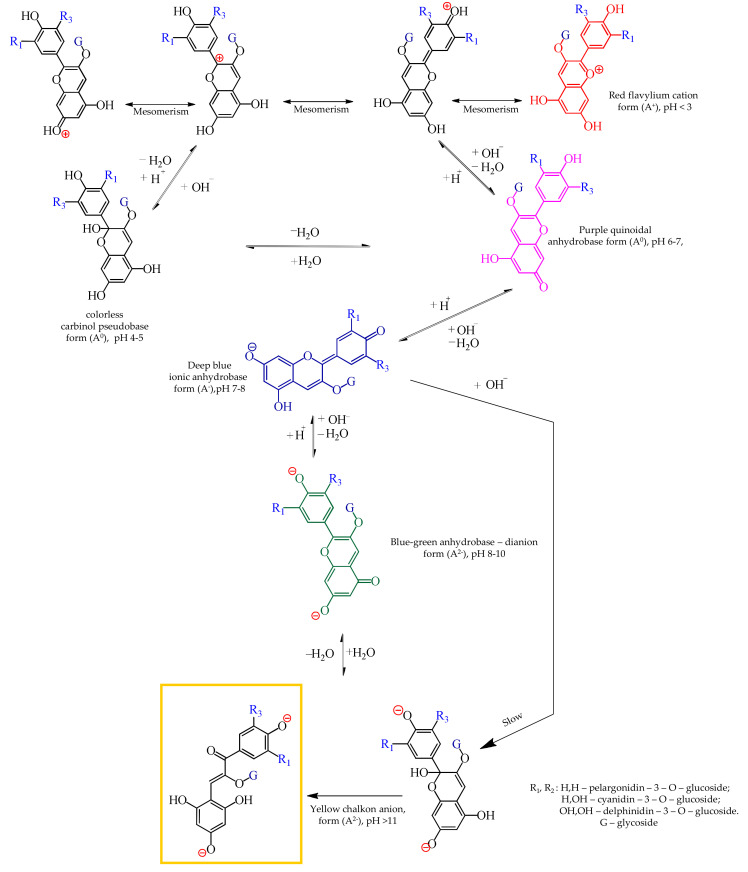
Structures of anthocyanins depending on the pH of the medium. R_1_, R_3_ are —H, or —OH, or —OCH_3_ groups; glycoside is glucose, rhamnose, arabinose, or galactose.

**Figure 2 ijms-21-07874-f002:**
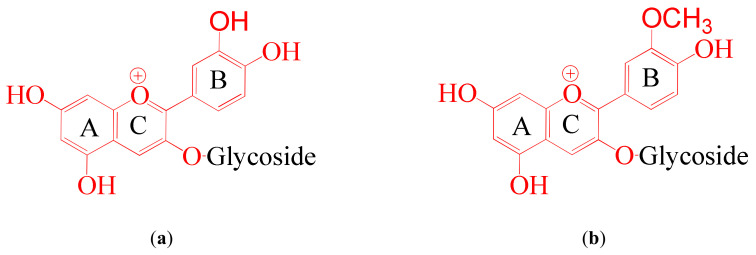
Structure of anthocyanins: cyanidin (**a**) and peonidin (**b**), contained in the external solution at pH 3. Redrawn from reference [1]. Glycoside is glucose, arabinose, or galactose.

**Figure 3 ijms-21-07874-f003:**
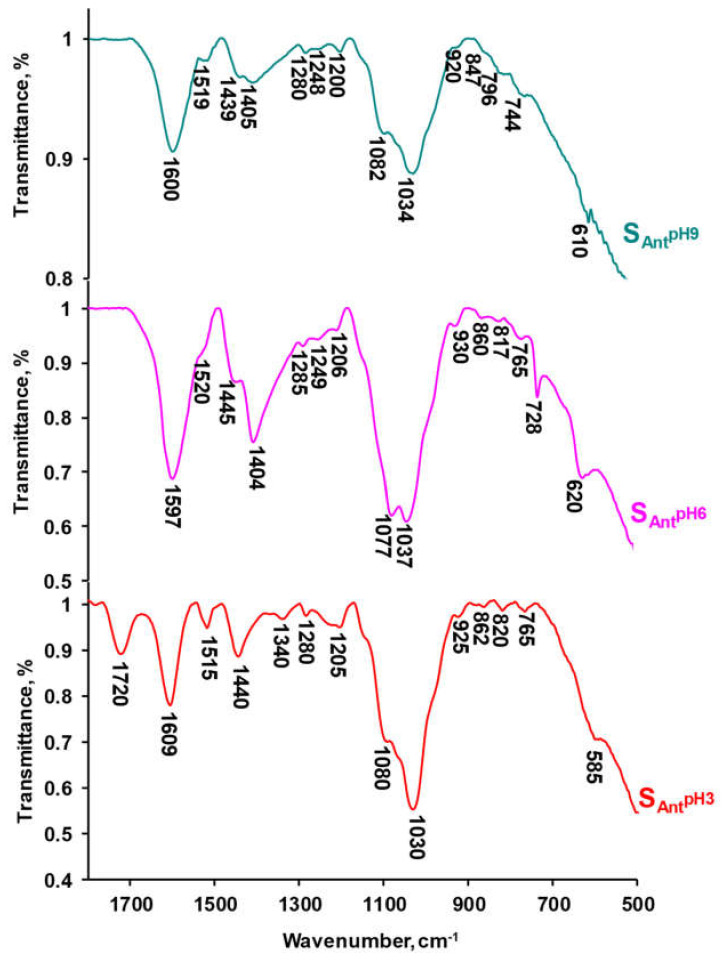
Effect of pH on the shape of the attenuated total reflection–Fourier transform infrared (ATR–FTIR) spectra in the anthocyanin fingerprint region (400–1800 cm^−1^).

**Figure 4 ijms-21-07874-f004:**
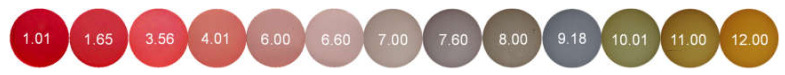
Effect of pH on the color of anthocyanin solutions. The pH values of the solutions are indicated for each color.

**Figure 5 ijms-21-07874-f005:**
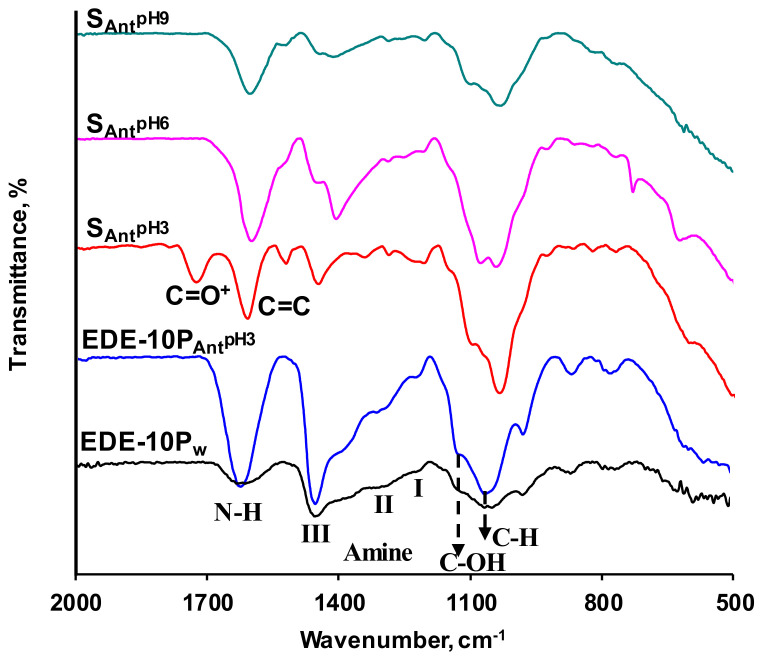
IR spectra of the EDE-10P samples equilibrated with deionized water (indicated by the subscript W) and an anthocyanin solution at pH 3 (indicated by the subscript _Ant_^pH3^). The IR spectra of the anthocyanin solutions at pH 3, 6, and 9 (indicated by the subscript S_Ant_^pHi^) are shown for comparison.

**Figure 6 ijms-21-07874-f006:**
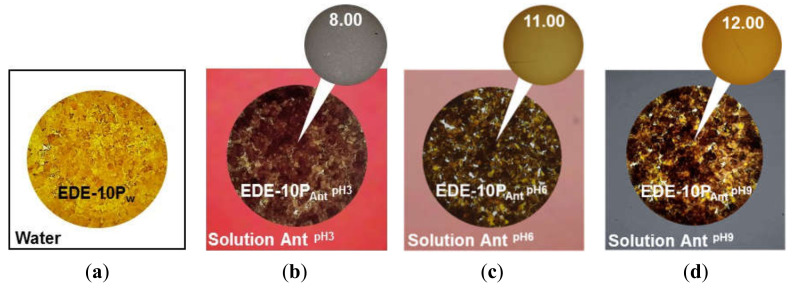
Color of the anion-exchange resin EDE-10P in the Cl^−^ form, equilibrated with deionized water (**a**) and model anthocyanin solutions at pH 3 (**b**), 6 (**c**), and 9 (**d**). The upper circles represent the closest anthocyanin color in solution on the colorimetric scale (Figure 4); the number inside the circle indicates the pH value corresponding to that color of the solution.

**Figure 7 ijms-21-07874-f007:**
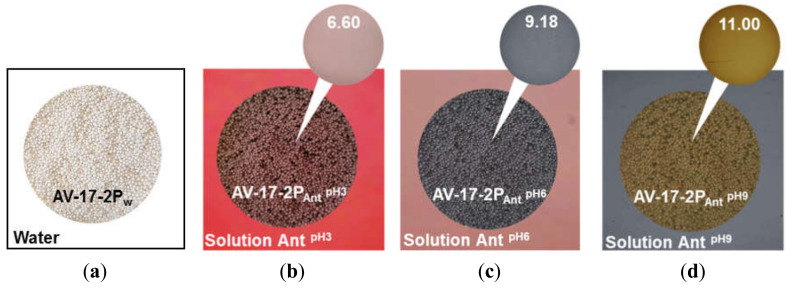
Color of the anion-exchange resin AV-17-2 in the Cl^−^ form, equilibrated with deionized water (**a**) and anthocyanin solutions at pH 3 (**b**), 6 (**c**), and 9 (**d**). The upper circles represent the closest anthocyanin color on the colorimetric scale in solution (Figure 4); the number inside the circle indicates the pH value corresponding to that color.

**Figure 8 ijms-21-07874-f008:**
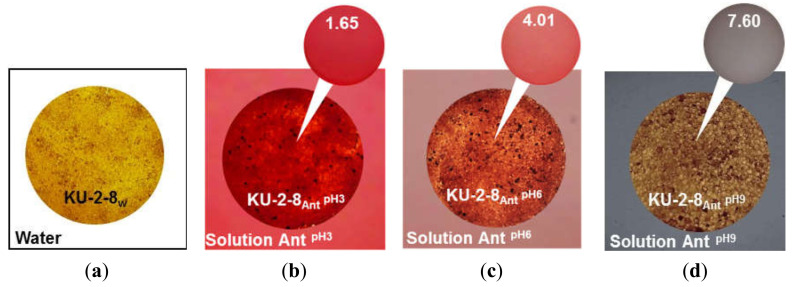
Color of the cation-exchange resin KU-2-8 in the Na^+^ form, equilibrated with deionized water (**a**) and anthocyanin solutions at pH 3 (**b**), 6 (**c**), and 9 (**d**). The upper circles represent the closest anthocyanin color on the colorimetric scale in solution (Figure 4); the number inside the circle indicates the pH value corresponding to that color.

**Figure 9 ijms-21-07874-f009:**
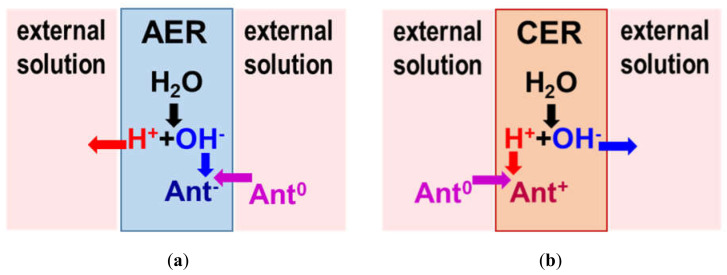
Schematic representation of the pH shift phenomenon in (**a**) anion-exchange resins (AERs), (**b**) cation-exchange resins (CERs), and external anthocyanin solutions (pH 6) surrounding the resins due to the Donnan exclusion of coions (H^+^ or OH^−^) that are the products of protonation/deprotonation reactions.

**Figure 10 ijms-21-07874-f010:**
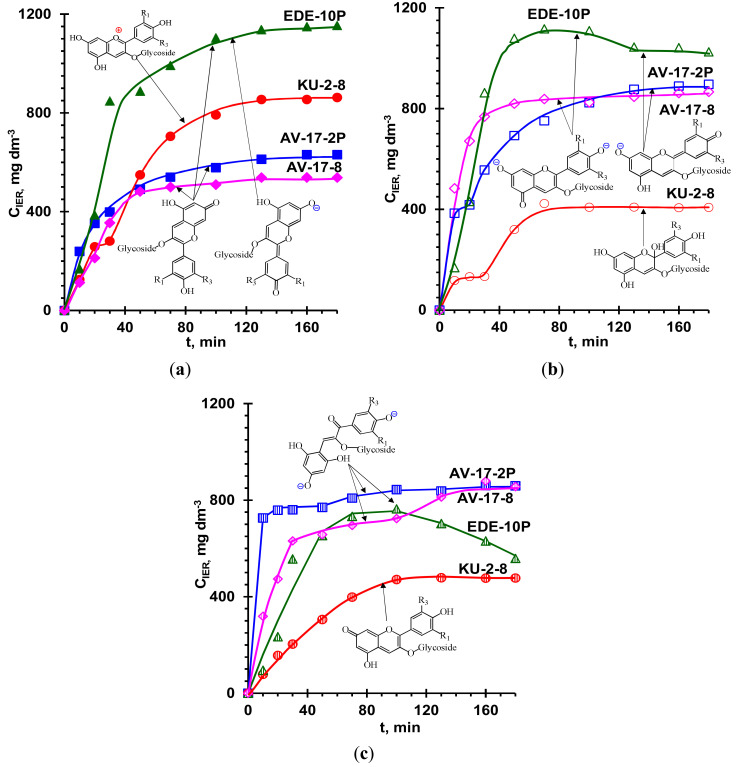
Kinetics of the anthocyanin adsorption by the ion-exchange resins from the aqueous solutions with initial anthocyanin concentration of 40 mg dm^−3^ and pH 3 (**a**), 6 (**b**), and 9 (**c**).

**Figure 11 ijms-21-07874-f011:**
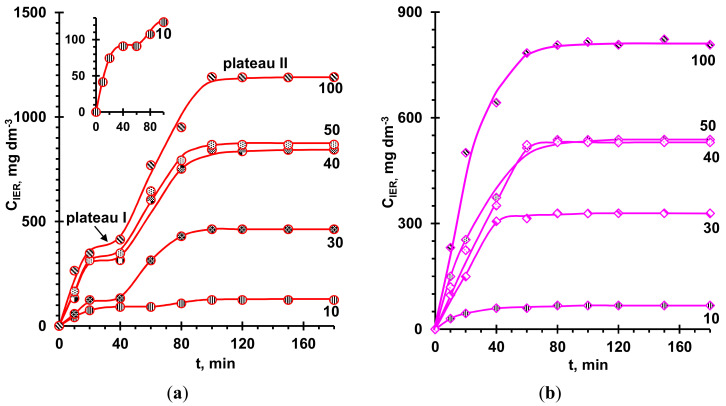
Kinetics of the anthocyanins adsorption by KU-2-8 (**a**) and AV-17-8 (**b**) ion-exchange resins from aqueous solutions with pH 3. The numbers near the curves indicate the concentration of anthocyanins in the solution, *C_S0_* (mg dm^−3^).

**Figure 12 ijms-21-07874-f012:**
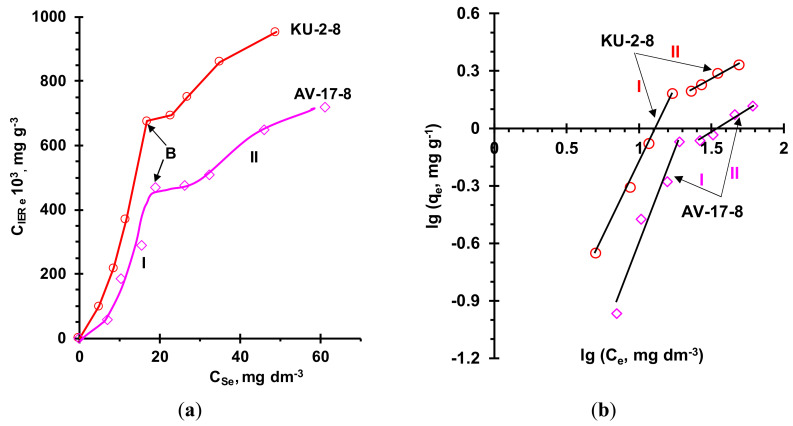
Equilibrium isotherms of anthocyanins adsorption (**a**) by resins KU-2-8 and AV-17-8 from anthocyanins solutions at pH 3 and the same isotherms processed using the Freundlich equation (Section 3.3.4.) (**b**). Point B corresponds to the transition from monomolecular to polymolecular adsorption.

**Table 1 ijms-21-07874-t001:** Composition of the anthocyanin extract from grape pulp.

Anthocyanin	Concentration, mg dm^−3^
Cyanidin-3-galactoside	105.6
Cyanidin-3-glucoside	20.8
Cyanidin-3-arabinoside	102.0
Peonidine-3-galactoside	51.8
Peonidine-3-glucoside	12.3
Peonidine-3-arabinoside	280.3
**Total concentration**	**572.8**

**Table 2 ijms-21-07874-t002:** Some characteristics of the ion-exchange resins determined in the present study.

IERs	Fixed Groups	Polymer Matrix	True Density, g cm−^3^	Water Content, g_H2O_ /g_wet_, % ^1^	Total Ion-Exchange Capacity, mmol cm^−3^
	Anion-exchange resins				
EDE-10P	=NH^+^,-NH_2_^+^, ≡N	ECH+PEPA	1.17 [82]	46	2.34 [83]
AV-17-2P	-N^+^(CH_3_)_3_	DVB2%+PS	1.06 ^1^	61	0.80 ± 0.02 [84]
AV-17-8	-N^+^(CH_3_)_3_	DVB8%+PS	1.13 [82]	45	1.12 ± 0.02 [84]
	Cation-exchange resin				
KU-2-8	-SO_3_^−^	DVB8%+PS	1.25 [82]	56	1.80 ± 0.01 [84]

^1^ This value is found by the method described in reference [85]. Abbreviations: polyethylene polyamine (PEPA), epichlorohydrin (ECH), divinylbenzene (DVB).

**Table 3 ijms-21-07874-t003:** Some characteristics of the systems under study.

	External Solution	KU-2-8	AV-17-8	AV-17-2P	EDE-10P
pH	3.00 ± 0.05	2.5 ± 0.5	7.0 ± 0.6	6.6 ± 0.4	8 ± 1
Anthocyanins’ electrical charge	1+	1+	0	0	0, 1– (a small amount)
**C_IER_/C_S0_* (t = 180 min) ^1^	-	20 ± 1	13 ± 1	15 ± 1	29
pH	6.00 ± 0.05	4.0 ± 0.5	8 ± 1	8.0 ± 0.5	10.5 ± 0.5
Anthocyanins’ electrical charge	0	0, 1+ (a small amount)	1–	1–	2–
**C_IER_/C_S0_* (t = 180 min) ^1^	-	10 ± 1	22 ± 1	22 ± 1	26 ± 1
pH	9.00 ± 0.05	7.6 ± 0.6	10.5 ± 1	11.0 ± 0.5	12 ± 1
Anthocyanins’ electrical charge	1–	0, 1– (a small amount)	2–	2–	2–
**C_IER_/C_S0_* (t = 180 min) ^1^	-	12 ± 1	22 ± 1	22 ± 1	14 ± 1

^1^ The adsorption factor of anthocyanins, achieved after 180 min of ion-exchange resins’ (IERs’) contact with an external solution, the concentration of anthocyanins in which *C_s0_* was 40 mg dm^−3^. The procedure for determining the concentration of anthocyanins in resins *C_IER_* is described in Section 3.3.1.

**Table 4 ijms-21-07874-t004:** Fitting results of experimental equilibrium isotherms using equations of Langmuir, Freundlich, and Brunauer–Emmett–Teller (BET) models (Section 3.3.4).

	KU-2-8				АВ-17-8			
	Langmuir (Section I)	Freundlich (Section I)	Freundlich (Section II)	BET	Langmuir (Section I)	Freundlich (Section I)	Freundlich (Section II)	BET
*R^2^*	0.947	0.998	0.982	0.884	0.922	0.961	0.982	0.844
*q_m_* (mg g_dry_^–1^)	−0.930	-		1.550	−0.251	-		0.850
*K_L_* (mg dm^–3^)^−1^	−0.040	-		-	−0.050	-		-
*K_F_* (mg dm^–3^)^−1^	-	0.002	0.420	-	-	0.003	0.154	-
*K_s_* (mg dm^–3^)^−1^	-	-		0.110	-	-		0.067
*K_B_* (mg dm^–3^)^−1^	-	-		0.008	-	-		0.008

**Table 5 ijms-21-07874-t005:** Composition of buffer solutions used to prepare colorimetric scales of anthocyanins.

Buffer Solution	pH
Potassium chloride (0.05 mol kg^−1^) + hydrochloric acid (0.097 mol kg^−1^)	1.01
Potassium tetraoxalate dihydrate (0.05 mol kg^−1^)	1.65
Potassium hydrogen tartrate (0.05 mol kg^−1^)	3.56
Potassium hydrogen phthalate (0.05 mol kg^−1^)	4.01
Potassium dihydrogen phosphate (0.26 mol kg^−^^1^) + Disodium hydrogen phosphate (0.04 mol kg^−1^)	7.00
Sodium tetraborate decahydrate (0.01 mol kg^−1^)	9.18
Sodium carbonate (0.025 mol kg^−1^) + sodium hydrogen carbonate (0.025 mol kg^−1^)	10.01
Sodium carbonate (0.05 mol kg^−1^) + sodium tetraborate decahydrate (0.00135 mol kg^−1^)	11.00
Potassium chloride (0.05 mol kg^−1^) + sodium hydroxide (0.012 mol kg^−1^)	12.00

**Table 6 ijms-21-07874-t006:** Lists of adsorption isotherms models under study.

Model	Non-Linear Form	Linear Form
Langmuir	$q_{e}=\frac{q_{m}K_{L}C_{e}}{1+K_{L}C_{e}}$	$\frac{1}{q_{e}}=\frac{1}{q_{m}K_{L}}\left(\frac{1}{C_{e}}\right)+\frac{1}{q_{m}}$
Freundlich	qe=KFCe1n	$\log q_{e}=\log K_{F}+\left(1/n\right)\left(\log C_{e}\right)$
BET	$q_{e}=\frac{q_{m}K_{S}C_{e}}{\left(1-K_{B}C_{e}\right)\left(1-K_{B}C_{e}+K_{S}C_{e}\right)}$	$\frac{C_{e}}{q_{e}\left(1-K_{B}C_{e}\right)}=\frac{1}{q_{m}K_{B}}+\frac{C_{e}\left(K_{B}-K_{S}\right)}{q_{m}K_{B}}$

where *q_e_* (mg g^−1^) is the amount of adsorbate adsorbed on the solid surface at equilibrium concentration *C_e_* (mg dm^−3^) and *q_m_* (mg g^−1^) is the amount of adsorbate corresponding to complete monolayer adsorption. *K_F_* (mg dm^−3^)^−1^ and *n* are Freundlich constants representing the Freundlich isotherm constant. *K_L_* (mg dm^−3^)^−1^ represents the Langmuir isotherm constant and *K_s_*, *K_B_* (mg dm^−3^)^−1^ are the equilibrium constants of adsorption for 1st and upper layers in BET isotherm, respectively.

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
