# Peer review of "Adsorption of Anthocyanins by Cation and Anion Exchange Resins with Aromatic and Aliphatic Polymer Matrices"

_ijms, 2020, doi:10.3390/ijms21217874_

Round 1

Reviewer 1 Report

The work from Pismenskaya et al. investigate the pH alteration in different IER during anthocyanines adsorption, studying color variations, adsorption kinetics and isotherms interpretations. The presented paper is well written, presenting a  sound experimental approach that leads to robust conclusions. It is opinion of this referee that this manuscript deserve to be pubblished on International Journal of Molecular Sciences.

Two minor suggestion, concerning the presented work:

- It is opinion of this referee that, considering the work performed on adsorption kinetics and isotherms, these terms can be added as key-words.

- In Paragraph 2.1.1., an extraction protocol from grape pulp is cited, but this referee was not able to find the experimental procedure in the following text. Some information are required.

Author Response

Dear Reviewer,

Thank you for your time and efforts. We greatly appreciate your very helpful comments and suggestions.

  1. It is opinion of this referee that, considering the work performed on adsorption kinetics and isotherms, these terms can be added as key-words.

Authors’ response:

Thanks for this suggestion. We added these terms as keywords in the revised manuscript.

Fixes are highlighted in yellow.

  1. In Paragraph 2.1.1., an extraction protocol from grape pulp is cited, but this referee was not able to find the experimental procedure in the following text. Some information are required.

Authors’ response:

Thanks for this remark. Indeed, an extraction protocol from grape pulp was not added because we used a ready-made extract of the company Frutarom Etol d.o.o. (https://www.iff.com/en/taste/frutarom), which does not provide information on the details of the extraction procedure.

We added the following phrase in Section 3.2:

“External solutions were prepared from an aqueous anthocyanins mixture. According to the manufacturer (DOO "Frutarom Etol", Skofja vas, Slovenia), it was extracted from grape pulp.”

Reviewer 2 Report

The study concerned the mechanisms that regulate the adsorption of anthocyanins on ion exchange resins.

Anthocyanin solutions at pH 3, 6 and 9 were prepared, then the anthocyanin solutions at different pH were characterized with their IR-spectra in the range between 400 and 1800 cm-1 which represents the fingerprint region of anthocyanis.

The correspondence between the pH of the solutions and their color when varying the pH has been described. Based on the color, the pH of the solutions inside the resins (internal solutions) in contact with the different aqueous solutions of anthocyanins was estimated and compared with the pH of the external anthocyanins solutions.

It was observed that when a cationic resin was used, the pH of the internal solution was 2 to 3 units lower than the pH of the external solution; on the contrary, when an anionic resin was used the pH of the internal solution was 2 to 3 units higher than the pH of the external solution.

These differences in pH helped to interpret the different kinetic adsoption isotherms of the studied resins.

The work is topical because it concerns the extraction processes of polyphenols which can be applied to different matrices, such as many by-products of the food industries (circular economy).

The work is well written, in particular the introduction and the discussion of the results are presented in clear and complete manner; I suggest its publication in the journal after minor revisions.

As regard what is reported at lines 496-499 “Point B on the “knee” of equilibrium isotherms characterizes the transition from monomolecular to polymolecular adsorption (section II). An indirect confirmation of the polymolecular adsorption of anthocyanins is the identification of dimers and trimers of these substances in ion-exchange membranes …”  I ask the Authors to clarify whether the presence of anthocyanin monomers adsorbed on the surface of the resins favors the subsequent adsorption of other anthocyanin monomers, for instance through p-p interactions, or rather the dimer and trimer forms are already present in the external anthocyanin solution.

Author Response

Dear Reviewer,

Thank you for your high appreciation of our work and very helpful suggestion. We tried to take into account this comment and made the necessary corrections to the manuscript.

  1. As regard what is reported at lines 496-499 “Point B on the “knee” of equilibrium isotherms characterizes the transition from monomolecular to polymolecular adsorption (section II). An indirect confirmation of the polymolecular adsorption of anthocyanins is the identification of dimers and trimers of these substances in ion-exchange membranes …” I ask the Authors to clarify whether the presence of anthocyanin monomers adsorbed on the surface of the resins favors the subsequent adsorption of other anthocyanin monomers, for instance through p-p interactions, or rather the dimer and trimer forms are already present in the external anthocyanin solution.

Authors’ response:

Thank you very much for this question. Here is the revised fragment of the text in page 16:

«Point B on the “knee” of equilibrium isotherms characterizes the transition from monomolecular to polymolecular adsorption (section II). The presence in the external solution of anthocyanin monomers adsorbed on the surface of the resins favors the subsequent adsorption of other anthocyanin monomers (due to π-π (stacking) interactions) as well as the dimer and trimer forms. An indirect confirmation of the polymolecular adsorption of anthocyanins is the identification of dimers and trimers of these substances in ion-exchange membranes [110] that had a chemical structure similar to the studied resins and put in contact with anthocyanin-containing solutions.»